# Induction of homologous recombination between sequence repeats by the activation induced cytidine deaminase (AID) protein

Jean-Marie Buerstedde[1]\*, Noel Lowndes[1], David G Schatz[2,3]

[1]Centre for Chromosome Biology, School of Natural Sciences, National University of Ireland, Galway, Galway, Ireland; [2]Department of Immunobiology, Yale University School of Medicine, New Haven, United States; [3]Howard Hughes Medical Institute, Yale University School of Medicine, New Haven, United States

**Abstract** The activation induced cytidine deaminase (AID) protein is known to initiate somatic hypermutation, gene conversion or switch recombination by cytidine deamination within the immunoglobulin loci. Using chromosomally integrated fluorescence reporter transgenes, we demonstrate a new recombinogenic activity of AID leading to intra- and intergenic deletions via homologous recombination of sequence repeats. Repeat recombination occurs at high frequencies even when the homologous sequences are hundreds of bases away from the positions of AID-mediated cytidine deamination, suggesting DNA end resection before strand invasion. Analysis of recombinants between homeologous repeats yielded evidence for heteroduplex formation and preferential migration of the Holliday junctions to the boundaries of sequence homology. These findings broaden the target and off-target mutagenic potential of AID and establish a novel system to study induced homologous recombination in vertebrate cells.

\*For correspondence: brstdd@
googlemail.com

**Competing interests:** The authors declare that no competing interests exist.

**Reviewing editor**: Michael R Botchan, University of California, Berkeley, United States

## Introduction

The activation induced cytidine deaminase (AID) is essential for all types of B cell-specific immuno-globulin (*Ig*) gene diversification—somatic hypermutation (SH), gene conversion (GC), and class switch recombination (CSR) (*Muramatsu et al., 2000*; *Revy et al., 2000*; *Arakawa et al., 2002*; *Harris et al., 2002*). AID likely initiates these processes by the deamination of deoxycytidines to uracils, as inactivation of the *DNA Uracil Glycosylase* (*UNG*) gene changed the SH spectrum at C/G bases toward transitions and impaired GC and CSR (*Di Noia and Neuberger, 2002*; *Rada et al., 2004*; *Saribasak et al., 2006*). Abasic sites resulting from the excision of AID-induced uracils are most likely removed by the apurinic/apyrimidinic endonuclease 1 (*Masani et al., 2013*), but how the nicked DNA strands are further processed to initiate alternatively CSR, GC, or SH is poorly understood.

AID-dependent double-strand breaks—believed to be generated by deamination of nearby cytidines on both strands of sequence repeats within switch regions—have been detected during CSR (*Wuerffel et al., 1997*; *Petersen et al., 2001*; *Rush et al., 2004*; *Schrader et al., 2005*). These breaks are normally joined by non-homologous end joining leading to the deletion of the intervening DNA sequence, but erroneous repair may lead to chromosomal translocations (reviewed by *Boboila et al., 2012*). It is uncertain, however, whether double-strand breaks routinely accompany SH, since breaks within hypermutating V segments were subsequently found to be AID independent (*Papavasiliou and Schatz, 2002*; *Bross and Jacobs, 2003*). Similarly, it remains unclear whether GC is initiated by a double strand-break or by a single-strand nick (*Yabuki et al., 2005*; *Nakahara et al., 2009*).

**eLife digest** Mutation can be harmful because changes to genes can disrupt vital processes or even cause diseases such as cancer. However, some genetic mutations can also be beneficial. Cells of the immune system, for example, need to create antibodies that attack a huge diversity of invading microbes. To do this, immune cells introduce changes into their genes to increase the diversity of the proteins that make up the antibodies.

An enzyme called AID is thought to play a crucial role in increasing the diversity of our antibodies by changing specific letters of the genetic code. Now, Buerstedde et al. have shown that the AID enzyme can also cause sections of DNA to be deleted from the genome.

Buerstedde et al. constructed pieces of DNA that include, in order, a gene that makes cells glow red, a gene that makes cells resistant to an antibiotic, and a gene that could make cells glow green. However, the very start of the 'green' gene was missing, which meant that it was switched off. Stretches of DNA were repeated in front of the 'red' and 'green' genes in some of the 'constructs'. After inserting this DNA into cells from chickens or mice, most cells glowed red, but some started to glow green instead. Green cells were killed by the antibiotic; and were only seen when cells carried the constructs with the repeating DNA. Cells that lacked the AID enzyme only glowed red, regardless of which DNA construct they carried.

Buerstedde et al. showed that AID causes the DNA constructs to align and re-arrange at the repeated sequences. As such, when the cells divide and their DNA is separated and packaged into newly formed cells, the DNA between the repeating sequences can be deleted. Thus, cells started to glow green because the 'on' switch at the start of the red gene ended up at the start of the green gene when the region in between was deleted. This also explains why green cells always died when exposed to the antibiotic, because this deletion removed the resistance gene too.

Buerstedde et al. suggest that when a cell attempts to correct the errors caused by the AID enzyme changing the letters in the DNA, it actually can trigger the exchange and deletion of repeated sequences. Future work is now needed to understand how this new role for the AID enzyme is regulated, and whether this role beneficial or harmful to the immune cells.

A number of studies have suggested that AID-mediated DNA lesions could be repaired by homologous recombination. Phosphorylated AID was reported to interact with Replication Protein A (RPA), a protein also required for recombination, which then recruited AID to single-stranded DNA and facilitated the deamination of cytidines (*Basu et al., 2005*). The RPA that accumulated at sites of AID-mediated DNA damage was subsequently shown to be associated with the RAD51 recombination protein (*Yamane et al., 2013*), suggesting the formation of homologous recombination intermediates. Notably, inhibition of homologous recombination decreased the viability of B cells expressing AID and induced widespread double-strand breaks and genomic instability (*Hasham et al., 2010*).

The chicken B-cell line DT40, easily modified by targeted gene integration (*Buerstedde and Takeda, 1991*), is a useful model to study AID-mediated GC and SH (*Di Noia and Neuberger, 2002*; *Arakawa and Buerstedde, 2009*). DT40 modifies its rearranged *Ig* light chain gene primarily by unidirectional gene conversion using nearby *pseudo-V* genes as conversion donor sequences (*Buerstedde et al., 1990*). However, GC decreases and SH increases if homologous recombination is impaired by inactivation of RAD51 paralogues (*Sale et al., 2001*), upon deletion of the upstream *pseudo V* genes that act as GC donor sequences (*Arakawa et al., 2004*) or following inactivation of the *UNG* gene (*Saribasak et al., 2006*). SH of the *Ig light chain* gene, as well as a green fluorescence protein (GFP) transgene were found to be strongly increased in the presence of a nearby Diversification Activator (DIVAC) sequence (*Blagodatski et al., 2009*). It was recently shown that this DIVAC consists of the chicken enhancer and enhancer-like elements and that it can be replaced by the *human Ig lambda* (*hIgλE*) and *Ig heavy intron* (*IgHiE*) enhancers (*Buerstedde et al., 2014*). CSR has been extensively studied using the murine B cell line CH12 (*Whitmore et al., 1991*; *Kinoshita et al., 1998*). This model system responds to the appropriate stimulation by increasing both Ig switch region transcription and AID expression (*Muramatsu et al., 2000*) and by recombining its endogenous Ig heavy chain locus or transfected switch region constructs by CSR.

While AID has been known to induce homologous recombination in the form of unidirectional gene conversion (*Arakawa et al., 2002*; *Harris et al., 2002*), there was previously no direct evidence that AID could mediate more complex rearrangements of sequence repeats. Using both DT40 and CH12 cells, we demonstrate here that AID can induce homologous recombination of repeats (RR) leading to frequent deletions between direct repeats.

## Results

### Single red fluorescence protein (RFP) reporters

Transfection of GFP-based hypermutation reporter constructs into DT40 suggested that in rare instances in which transgenes had undergone multi-copy integration there was frequent copy number contraction during cell clone expansion (data not shown). To investigate this unexpected form of genomic instability in AID expressing cells, we decided to develop dual color reporters to reveal transgene recombination by changes in cellular fluorescence.

As initial controls, single red fluorescence protein (RFP) reporters similar to the previously described *GFP2* (*Blagodatski et al., 2009*) were made encoding either the *tdTomato* (*tdT*) or *DS-Red Express* (*DsR*) gene (*Figure 1A*). The *RFP* genes were efficiently translated due to the presence of two in-frame ATG start codons, one at the beginning of the 5′ untranslated exon and one at the beginning of the *RFP* open reading frame (marked by arrows in *Figure 1A*). These constructs as well as all others used in our study were integrated in a targeted manner into the IgL(−) DT40 cell line at the position of the deleted *IgL* locus (*Blagodatski et al., 2009*). FACS analysis of subclones of the transfectants revealed a trailing cloud of cells with decreased red fluorescence (*Figure 1B,C*). Red fluorescence loss was stimulated 10- to 30-fold in the presence of the human *Ig lambda* enhancer (*hIgλE*) DIVAC (*Buerstedde et al., 2014*) (compare *Figure 1B2* to *Figure 1B3*, *Figure 1B4* to *Figure 1B5*; and *Figure 1C*). These results, combined with our previous extensive analyses of *GFP*-based reporter constructs (*Blagodatski et al., 2009*; *Buerstedde et al., 2014*), suggested that the loss of red fluorescence was due to inactivation of the *RFP* transgenes by AID dependent, DIVAC-stimulated hypermutation events. Interestingly, a discrete cell population of intermediate red fluorescence was seen in transfectants of the *tdT* but not the *DsR* gene constructs (*Figure 1B2* and *Figure 1B3*, circled). Since tdT is encoded by a direct tandem repeat of a *RFP* gene sequence (*Shaner et al., 2004*), the intermediate red fluorescence possibly reflected the loss of one of the *tdT* repeats by intragenic recombination.

### Activation of green fluorescence in dual color gene constructs

We then proceeded to reporters containing two fluorescent reporter genes: an upstream *tdT* which was efficiently translated due to the presence of the same in-frame ATG start codons as in the single RFP control constructs (marked by arrows in *Figure 2A*), and a downstream *GFP* which lacked an ATG start codon. For simplicity and consistency, the names of the dual color reporter constructs reflected only the *RFP* gene (either *tdT* or *DsR*), they contained and the presence of DIVACs and sequence repeats. In the first of these constructs, DIVAC_tdT, the upstream *tdT* and the downstream *GFP* genes shared no sequence homology and were driven by different promoters, *RSV* and *Ubiquitin C*, respectively (*Figure 2A*). Transfectants of the DIVAC_tdT construct gave rise to cell populations of decreased red fluorescence in the R3 gate and of intermediate fluorescence (*Figure 2B1*, *Figure 2C*) identical to the ones seen for single *tdT* gene transfectants (*Figure 1B3*, *Figure 1C*).

To test whether the presence of sequence repeats induced instability, a 344 bp direct repeat sequence (*iHS*) was inserted into the introns of both the *tdT* and the *GFP* genes yielding the DIVAC_iHS_tdT_iHS construct (*Figure 2A*). Subclones of transfectants showed, in addition to cells in the R3 gate (hereafter, R3 cells), green fluorescence positive, red fluorescence negative cells in the R1 gate at median frequencies of about 0.8% (*Figure 2B2*, *Figure 2C*). Cells of this phenotype were expected to arise if homologous recombination between the *iHS* sequences deleted the *tdT* gene and activated *GFP* translation due to the gain of the first ATG start codon previously located upstream of *tdT*.

In other constructs, both the *tdT* and *GFP* genes were driven by *RSV* promoters and they either lacked (RSV_tdT_RSV) or contained (DIVAC_RSV_tdT_RSV and DIVAC2_RSV_tdT_RSV) a DIVAC element (*Figure 2A*). Surprisingly, subclones gave rise to red fluorescence negative cells of intermediate green fluorescence in the R2 gate (note that homologous recombination between the *RSV* promoters does not provide *GFP* with a start codon and hence would not be expected to yield cells with strong GFP fluorescence in the R1 gate). The appearance of these cells was strongly enhanced

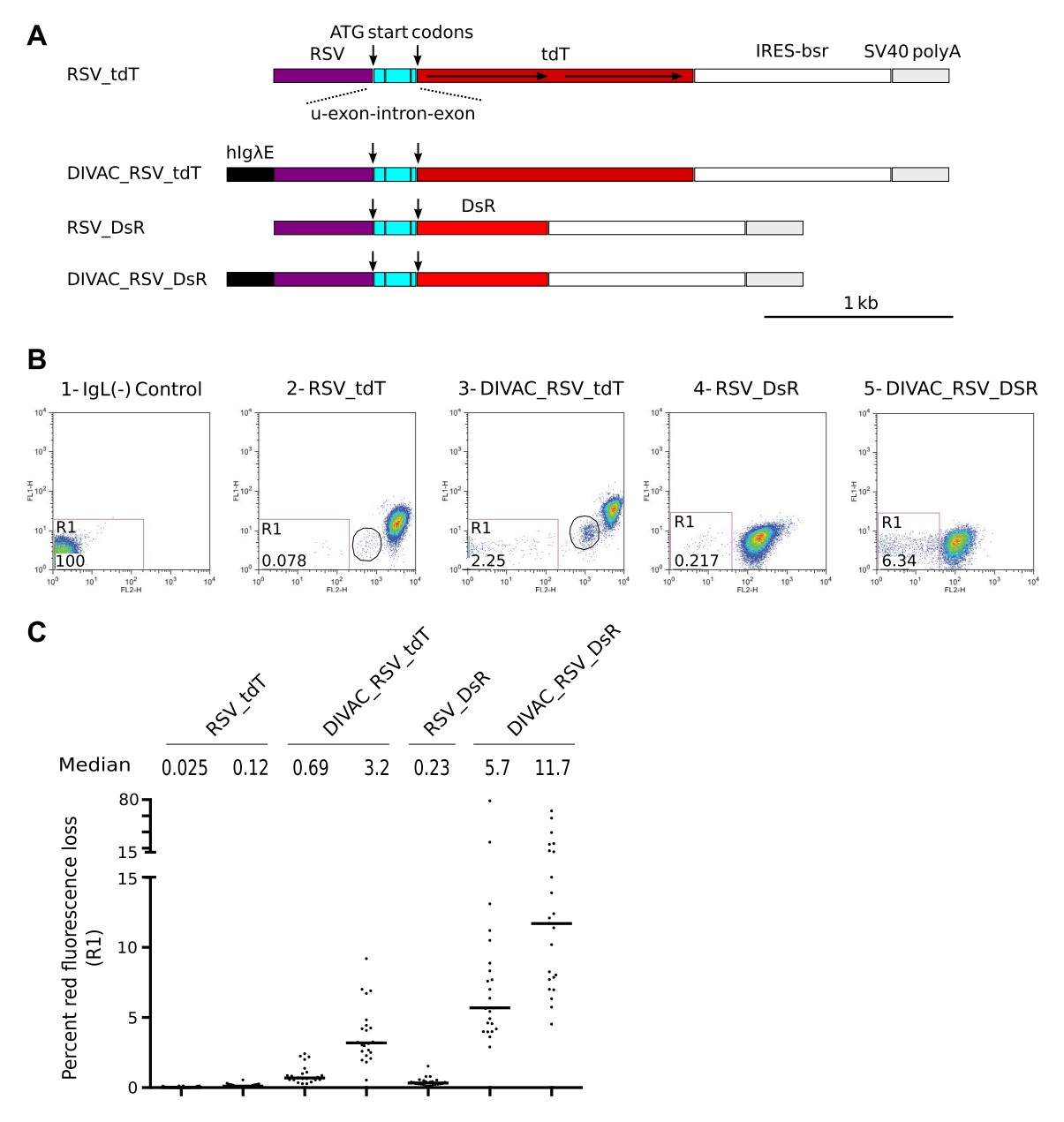

Figure 1. FACS analysis of DT40 transfectants expressing red fluorescence reporter constructs. (**A**) Diagrams of the constructs. Sequences important for the behavior of the construct are labeled and color coded: *hIgλE*—human *Igλ* enhancer; *RSV*—*rous sarcoma virus* promoter; u-exon-intron-exon—upstream splice cassette; *tdT* and *DsR*–*Tomato* and *DsRed* open reading frames, respectively; *IRES-bsr*–internal ribosome entry site followed by the *blasticidin resistance* open reading frame; *SV40* polyA—*SV40* polyadenylation signal. The names of the constructs indicate the presence of DIVACs, the promoter, the fluorescence genes and sequence repeats. The tandem repeats of *tdT* are marked by lines with arrows. (**B**) Two color FACS dot plots of the non-transfected IgL(−) cell line and representative subclones derived from primary transfectants. The levels of green and red fluorescence are plotted according to the x-axis (FL1) and y-axis (FL2), respectively. The number of the plot and the name of the transfected construct are indicated above. Gates and the percentage of gated cells are indicated within the plots. Populations of intermediate red fluorescence are circled. (**C**) Graphs showing the percentages of gated cells for all subclones of each independent transfectant. The median percentage of gated cells is indicated by the bar and numerically displayed above the graph for each transfectant.

in the presence of either the *hIgλE* or *hIgHiE* DIVAC (compare *Figure 2B4* to *Figure 2B5,B7*; and *Figure 2C*). Removal of the AID expression cassette from the DIVAC_iHS_tdT_iHS and DIVAC_RSV_tdT_RSV transfectants resulted in stable red fluorescence only expression (*Figure 2B3, 2B6*, *Figure 2C*)

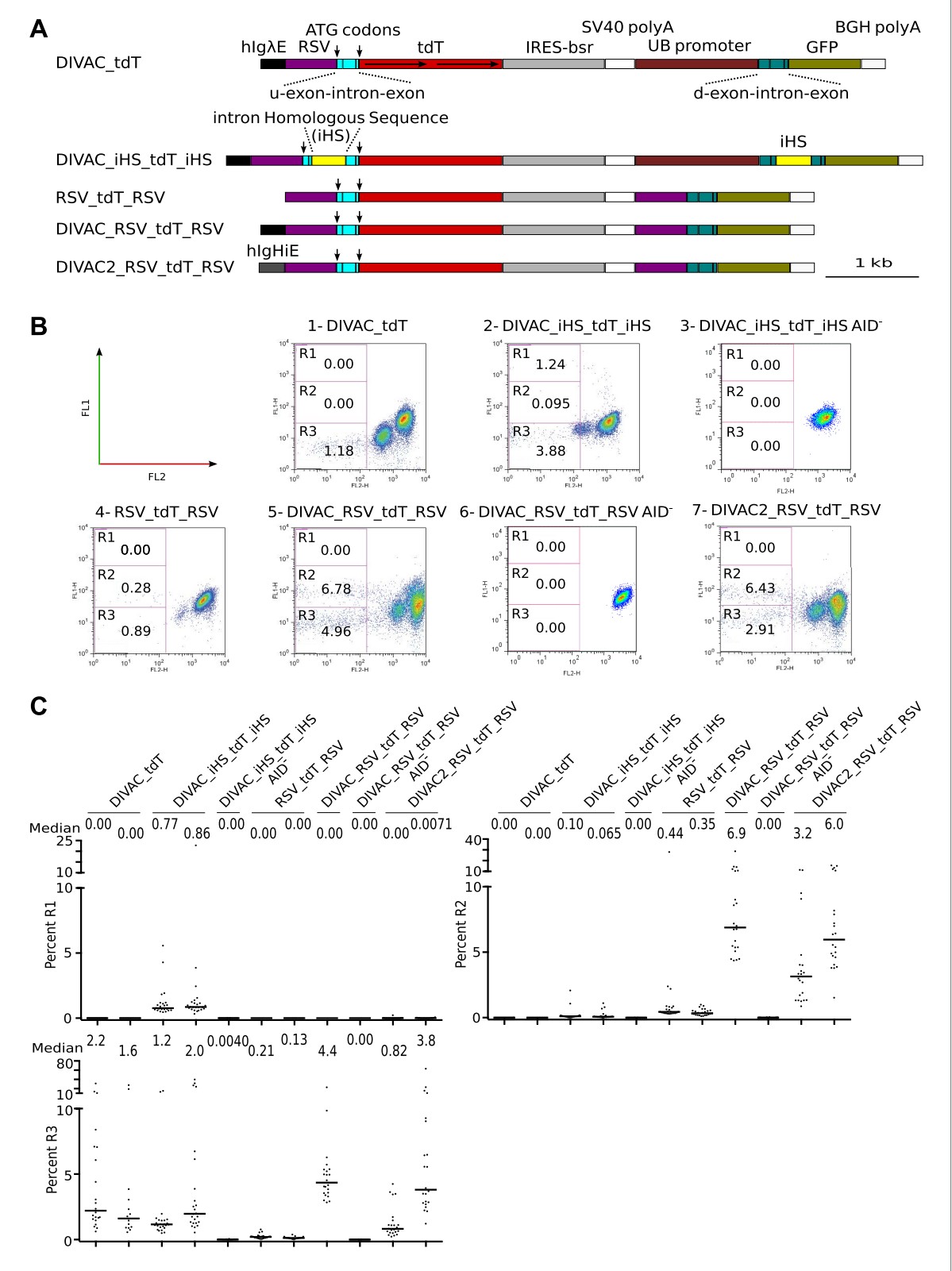

**Figure 2**. FACS analysis of DT40 transfectants expressing dual *tdT*/*GFP* fluorescence reporter constructs. (**A**) Diagrams of the constructs. Sequences important for the behavior of the construct are labeled and color coded as in *Figure 1* and explained in the following: *hIgHiE*—human *Ig heavy chain intron* enhancer; *UB* promoter—human *Ubiquitin C* promoter; d-exon-intron-exon—downstream splice cassette; *GFP*—*GFP* open reading frame;
*Figure 2. Continued on next page*

*Figure 2. Continued*

*BGH* polyA—*Bovine Growth Hormone* polyadenylation signal. The names of the dual fluorescence constructs indicate only the presence of DIVACs, the *RFP* gene and sequence repeats. (**B**) FACS dot plots of representative subclones derived from primary transfectants. The number of the plot and the name of the transfected construct are indicated above. Plots of subclones in which the AID expression cassette have been deleted are labeled AID⁻. Gates and the percentage of gated cells are indicated within the plots. (**C**) Graphs for each gate showing the percentages of gated cells for all subclones of each independent transfectant. The median percentage of gated cells is indicated by the bar and numerically displayed above the graph for each transfectant.

indicating that all fluorescence variation depended on AID. Similar results were obtained after transfection of constructs containing the *DsR* gene instead of the *tdT* gene (*Figure 3A–C*).

## Green fluorescence positive cells are due to intergenic deletions

The *blasticidin resistance* gene (*bsr*), which is positioned between the *RFP* and *GFP* genes (*Figure 2A*), would be lost if the *iHS* or the *RSV* promoter repeats of the dual fluorescence constructs underwent homologous recombination of repeats (RR). Addition of blasticidin to the culture indeed eliminated R1 cells from the DIVAC_iHS_tdT_iHS transfectant and strongly reduced R2 cells of *RSV* repeat transfectants (*Figure 4A*, compare upper to lower panel) consistent with the deletion of *bsr* in R1 and R2 cells. In contrast, R3 cells and cells of intermediate red fluorescence remained blasticidin resistant (*Figure 4A*, lower panel), as expected if these events reflected inactivation of the *RFP* gene by hypermutation and intragenic recombination of the *tdT* repeats, respectively.

Subclones derived from cells showing altered fluorescence continued to generate variant cell populations during expansion (*Figure 4B*). A subclone derived from a cell of intermediate red fluorescence still generated R2 and R3 cells presumably due to ongoing *RSV* repeat recombination and hypermutation of the recombined *tdT* gene respectively (*Figure 4B1*). R2 subclones still generated R3 cells presumably due to deleterious hypermutation of the rearranged *GFP* gene (*Figure 4B2 and 2B3*), and a R3 subclone still generated R2 cells as expected for ongoing recombination of the *RSV* repeat (*Figure 4B4*).

To detect rearrangements of the constructs, genomic DNA of a DIVAC_RSV_tdT_RSV primary transfectant as well as subclones derived thereof from an intermediate red fluorescence cell or an R2 cell was amplified by PCR using various combinations of primers (*Figure 4C*). Whereas the 1/3 primer pair amplified two fragments of about 1.7 kb and 2.5 kb from the primary transfectant (*Figure 4C*, lane 2 marked by asterisks), amplification from the intermediate red fluorescence subclone yielded only the lower fragment consistent with the deletion of one of the *tdT* repeats in cells of intermediate fluorescence (*Figure 4C*, lane 6). The 1/2 and 1/3 primer pairs did not amplify the DNA of the R2 subclone (*Figure 4C*, lane 9 and 10) and the 1/4 and 1/5 primer pairs amplified only fragments of about 1.1 kb and 1.9 kb (*Figure 4C*, lane 11 and 12), but not the larger fragments of 4–5 kb size seen in addition to the lower size fragments in amplifications from the primary transfectant (*Figure 4C*, lane 3 and 4) and the intermediate red fluorescence subclone (*Figure 4C*, lanes 7 and 8). This was consistent with a large deletion that included one *RSV* repeat and the intervening sequences in R2 cells.

The 1.9-kb fragment amplified by primer pair 1/5 from sorted DIVAC_RSV_tdT_RSV R2 cells was subcloned and sequenced. 12 sequences showed deletions of one *RSV* repeat and the intervening sequence with no additional nucleotide changes when compared to the sequence of the transfected construct (data not shown). These analyses demonstrate that AID induces intragenic homologous recombination between the *tdT* repeats and intergenic recombination between the *RSV* promoter sequences at high frequencies. However, as the deletion in R2 cells included the ATG codons upstream of the *tdT* gene, the reason for the increased green fluorescence of R2 cells remains unclear. It might be related to enhanced *GFP* transcription or the gain of an alternative *GFP* translation start codon in the recombinants.

## Recombination between homeologous sequences

To better understand AID-mediated deletions, we made the DIVAC_uHS_tdT_dHS construct (*Figure 5A*) in which an upstream (*uHS*) and downstream (*dHS*) homeologous sequence differed about every 50 base pairs by single nucleotide substitutions, deletions, or insertions. The duplicated sequence consisted of the *RSV* promoter and a 560-bp sequence encompassing the first exon, the intron, and the beginning of the second exon. As only *uHS* provided in-frame ATG start codons, the unmutated

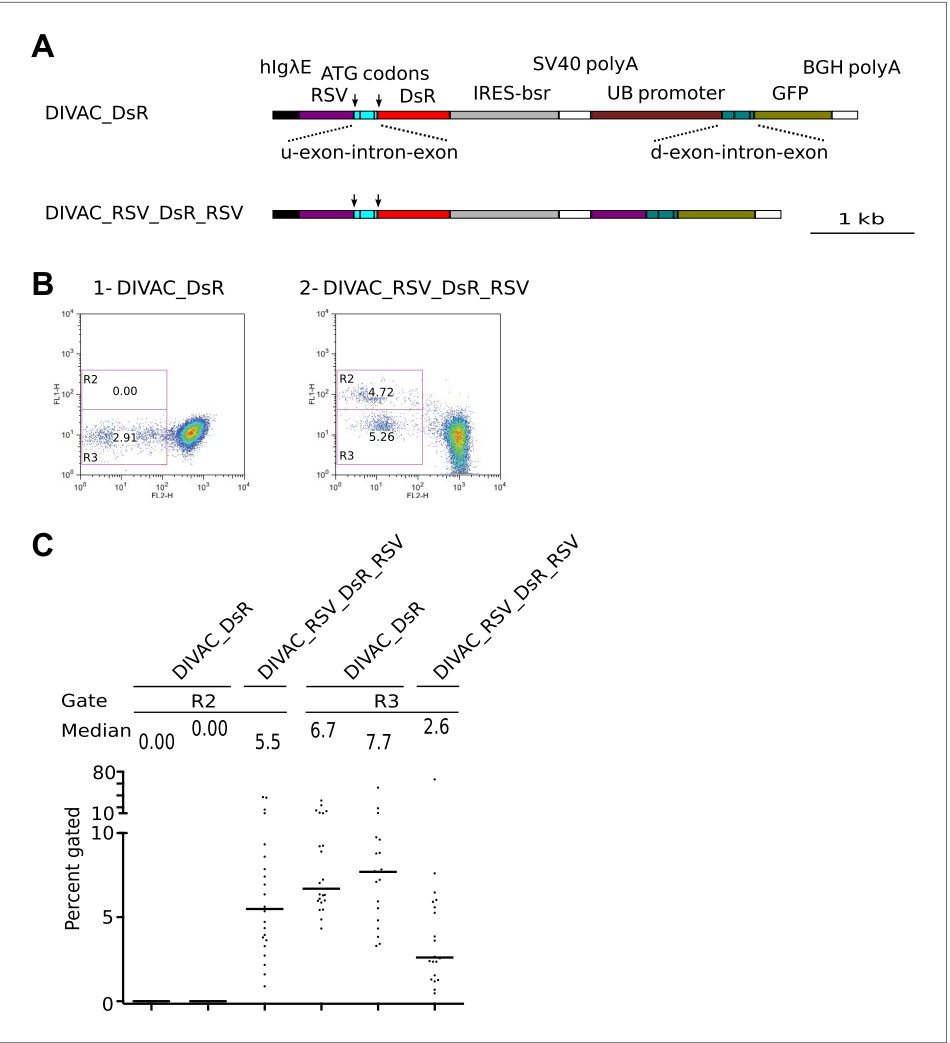

**Figure 3**. FACS analysis of DT40 transfectants expressing dual *DsR/GFP* fluorescence reporter constructs. (**A**) Maps of the constructs. (**B**) FACS dot plots of representative subclones derived from primary transfectants. (**C**) Graphs for each gate showing the percentages of gated cells for all subclones of each independent transfectant. The median percentage of gated cells is indicated by the bar and numerically displayed above the graph for each transfectant.

form of the construct supported tdT but not GFP expression. However, recombination events accompanied by a crossover downstream of the first ATG start codon would place this codon upstream of the *GFP* gene, causing loss of red fluorescence and gain of green fluorescence.

DIVAC_uHS_tdT_dHS transfectants generated cells in the R1, R2, and R3 gates, as well as cells of intermediate red fluorescence in the R4 gate (*Figure 5B,C*) as expected from the behavior of previous constructs. The median percentage of R1 and R2 events was lower, however (*Figure 5B,C*), most likely due to inhibition of homologous recombination by the heterologies of the *uHS/dHS* sequences. At least two other minor populations of increased green or red fluorescence were seen in the R5 and R6 gates, respectively (*Figure 5B,C*).

A 6-week culture of a primary DIVAC_uHS_tdT_dHS transfectant was sorted for cells in the R1–R6 gates (*Figure 6A*), and the sorted populations were subcloned. Only subclones derived from R1- and R2-gated cells were killed upon addition of blasticidin, indicating that these cells but not the cells in other gates had deleted the *bsr* gene (data not shown). Subclones derived from gated cells in turn generated subpopulations with characteristic patterns of fluorescence. Subclones derived from R1 (*Figure 6B1*) and R2 (*Figure 6B2*) cells produced double negative R3 cells presumably due to the inactivation of the rearranged *GFP* gene by hypermutation. Subclones derived from double negative

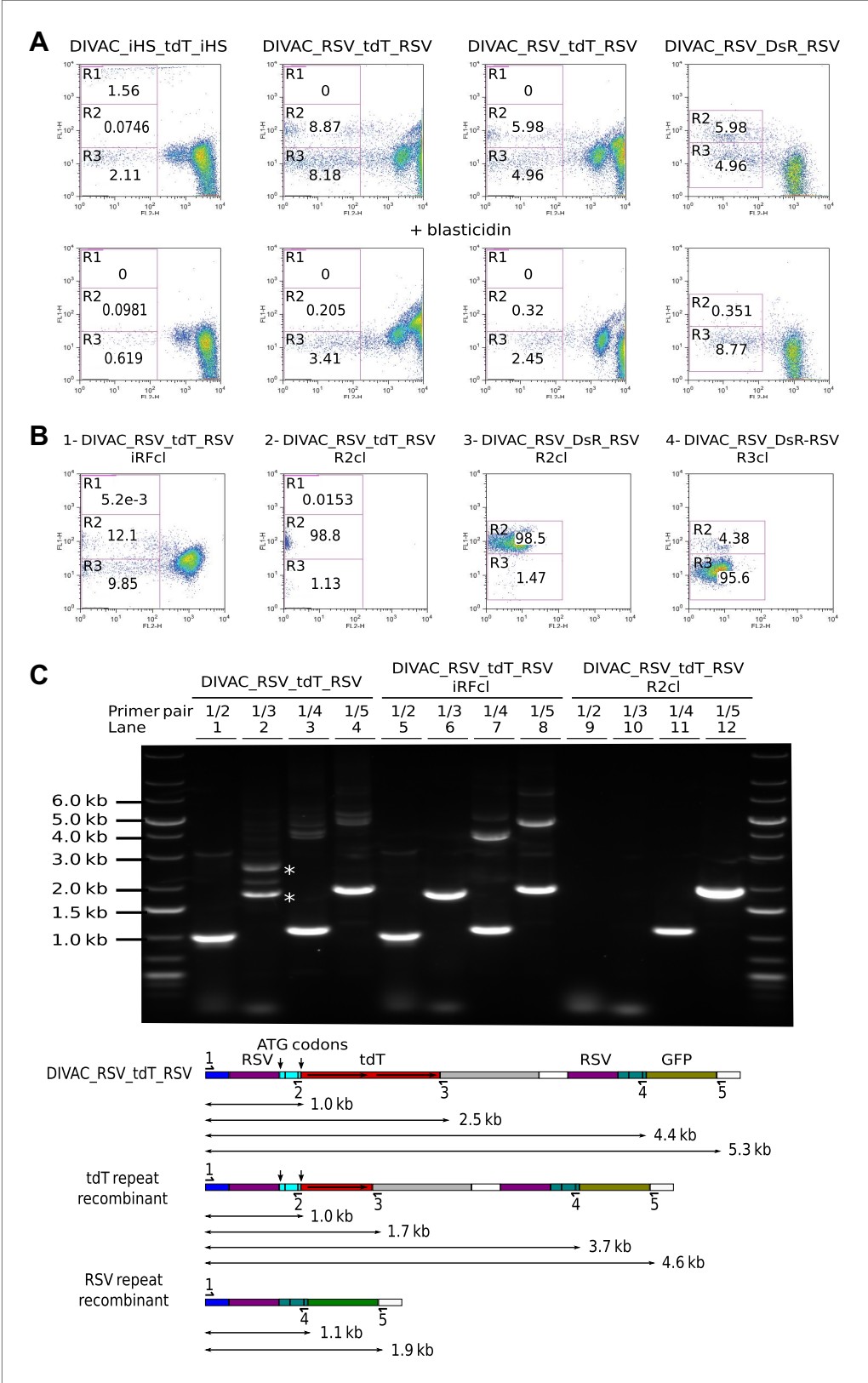

**Figure 4.** Homologous recombination of sequence repeats in gated cells. (**A**) FACS profiles of primary transfect-ants 20 days after transfection. Cells analyzed in the upper row were cultured in the absence of blasticidin, those in *Figure 4. Continued on next page*

*Figure 4. Continued*

the lower row in the presence of blasticidin. (**B**) Representative FACS profiles of subclones derived from a cell of intermediate red fluorescence or from R2- and R3-gated cells. The transfected constructs are indicated above the profiles by name, with the origin of the precursor cell indicated by the suffix. (**C**) Top: agarose gel electrophoresis of PCR products amplified from DNA of the DIVAC_RSV_tdT_RSV transfectant and two of its subclones. The first subclone is derived from a cell of intermediate red fluorescence, the second from a R2 cell. The primers used are indicated above the numbered lanes. Bands representing amplifications of the un-rearranged and rearranged *tdT* genes are marked by asterisks in lane 2. Below: the diagram shows the positions of the primers and the expected sizes of the PCR products for the transfected construct and its recombinants. The increased GFP expression of recombinants is indicated by color changes of the rectangle representing the *GFP* open reading frame.

R3 cells (*Figure 6B3*) gave rise to green fluorescence positive cells, presumably due to *uHS/dHS* repeat recombination, but also to cells having regained red fluorescence presumably due to repair of their mutated *tdT* gene. R5 subclones generated R2 and R3 cells, as well as cells of intermediate red fluorescence relative to R5 cells, but no events in the R1 gate (*Figure 6B5*). Finally, R6 subclones produced a FACS profile similar to that of the DIVAC_uHS_tdT_dHS primary transfectant with the difference that all cell populations displayed increased green fluorescence (*Figure 6B6* and data not shown).

Genomic DNA of two subclones of each gated population was analyzed by PCR (*Figure 6C*). While the 1/3 primer pair did not amplify DNA from R1 (*Figure 6C*, lane 1 and 3) and R2 (*Figure 6C*, lane 5 and 7) subclones, the 1/5 primer pair produced only a single fragment of about 2.3 kb as expected for *uDS/dHS* recombination in the precursor cell of the R1 and R2 subclones (*Figure 6C*, lane 2, 4, 6 and 8). Amplification from R4 subclones by the 1/3 primer pair produced a single fragment of about 2.2 kb (marked by an asterisk in *Figure 6C*, lane 13 and 15) consistent with the deletion of one *tdT* repeat in R4 cells. PCR amplifications from the R3, R5, and R6 subclones produced band patterns similar to one another, showing fragments of the size expected for cells carrying the non-rearranged construct as well as smaller fragments expected from cells having undergone either *tdT* (detected by the 1/3 primer pair) or *uHS/dHS* repeat (detected by 1/5 primer pair) recombination during subclone expansion (*Figure 6C*, lanes 9–12 and 17–24). This indicated that R1 and R2 cells had recombined *uHS/dHS*, and R4 cells had recombined the *tdT* repeat, whereas R3, R5, R6 cells did not carry detectable rearrangements.

## Mapping of crossing-overs and evidence for heteroduplex formation

If the deletions in R1 and R2 cells occurred by homologous recombination, the nucleotide differences distributed along the entire length of *uHS* and *dHS* (*Figure 7A*) would allow the mapping of crossover sites into an interval between two polymorphic positions. Analysis of recombined sequences amplified by the 1/5 primer pair from gated R1 and R2 cells indeed provided evidence for variable crossover positions (*Figure 7B*). All R1 sequences had maintained *uHS*-specific sequence at least until the single nucleotide insertion polymorphism (marked by two vertical arrows in *Figure 7A*) that puts the upstream ATG codon in frame with *tdT* prior to recombination and in frame with *GFP* after recombination. This explains the strong increase in green fluorescence seen in R1 cells. In contrast, all R2 sequences maintained *dHS*-specific sequence at this position (*Figure 7B*) suggesting again that modest increase of green fluorescence in R2 cells is either due to enhanced *GFP* transcription or the gain of an alternative *GFP* translation start codon.

Five R1 sequences (S1, S2, S3, S4, and S7) maintained *uHS* sequence down to the last polymorphic nucleotide in front of the *GFP* reading frame and five R2 sequences (S14, S20, S23, S26, S30) maintained *dHS* sequence up to the last polymorphic nucleotide downstream of the *hIgλE* sequence, suggesting preferential crossover events near the boundaries of sequence homology. Interestingly, six sequences from R2 cells (S9, S10, S12, S13, S18, S21) contained either single or multiple *dHS*-specific nucleotides upstream of *uHS*-specific nucleotides (marked by red circles in these sequence in *Figure 7B*), most easily explained by mismatch repair of recombination associated heteroduplexes.

Four sequences amplified from gated R4 cells by the 1/3 primer pair showed deletion of one *tdT* repeat and the 69-bp intervening sequence between *tdT* repeats with no other sequence changes confirming that cells of intermediate red fluorescence were due to intragenic *tdT* repeat recombination (data not shown).

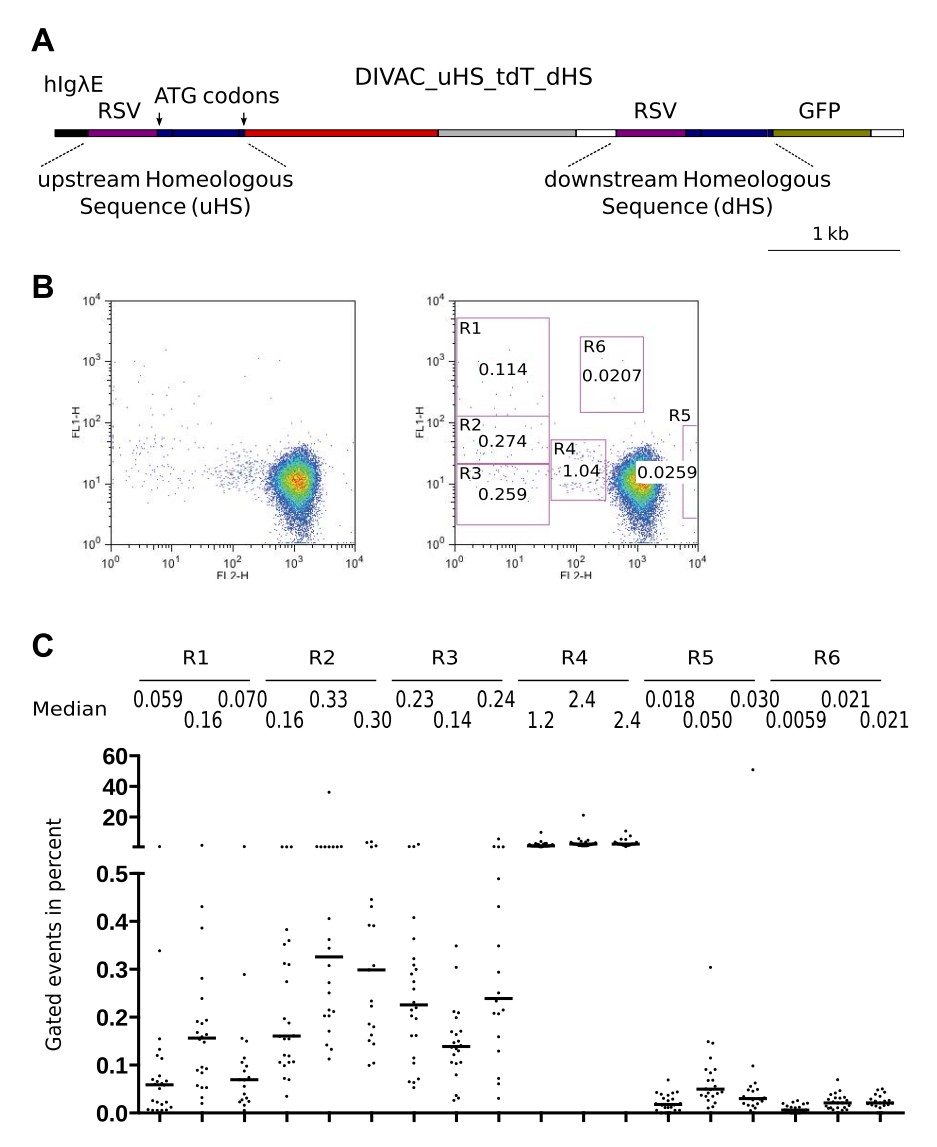

**Figure 5**. FACS profiles of transfectants of the DIVAC_uHS_tdT_dHS construct. (**A**) Diagram of the construct. The upstream and downstream homeologous sequence repeats (*uHS* and *dHS*, respectively) are highlighted. (**B**) FACS profile of a representative subclone. (**C**) Graphs showing the percentages of gated cells for subclones of three independent transfectants. The median percentage of gated cells of all subclones is indicated by the bar and numerically displayed above the graph for each transfectant.

## Gene conversion events giving rise to R5 and R6 cells

Two *uHS* sequences amplified by the 1/3 primer pair from R5 subclones showed that the middle part had converted to *dHS*-specific sequence thereby erasing the first ATG start codon (*Figure 7C*). If the canonical tdT protein translated from the second ATG codon were more active than the artificial tdT fusion protein translated from the first ATG codon, this would explain the increased red fluorescence of R5 cells. The loss of the first ATG start codon is also consistent with the observation that R5 subclones were unable to generate R1 cells. The observed sequence changes likely reflected unidirectional *dHS*-templated gene conversions, as an increase of green fluorescence was not observed in R5 cells, but would have been expected if a reciprocal exchange had introduced the first ATG codon into *dHS*.

Surprisingly, *dHS* sequences amplified from an R6 subclone (with very high green fluorescence) by the 1/5 primer pair showed the insertion of seven *uHS* specific nucleotides including the downstream

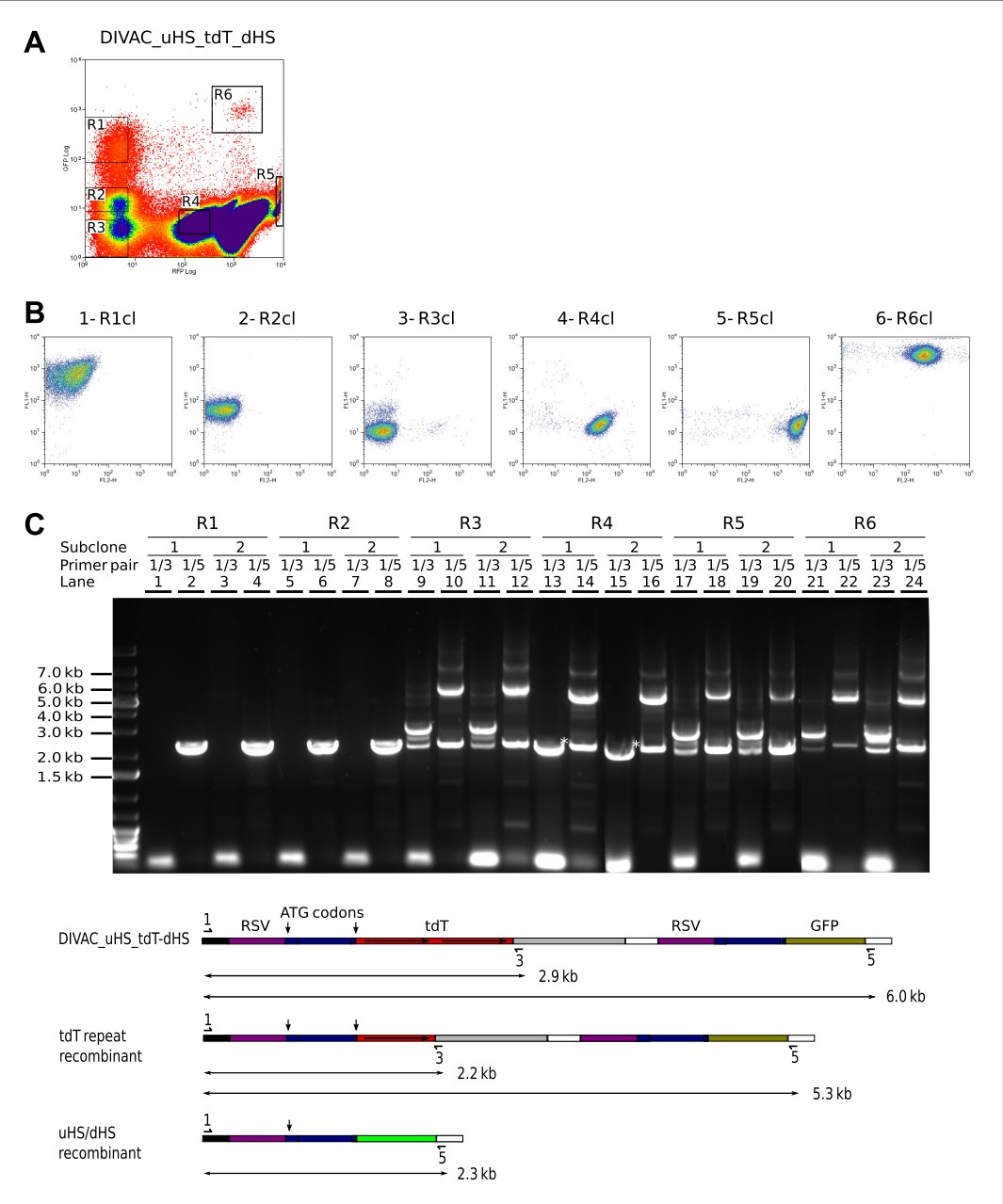

**Figure 6**. Recombination of the uHS/dHS repeat in R1 and R2 cells of the DIVAC_uHS_tdT_dHS transfectant.
(**A**) FACS profile of a DIVAC_uHS_tdT_dHS transfectant showing the gates used for preparative sorts. (**B**) FACS
plots of representative subclones. The gate from which the precursor cell of the subclone is derived is shown above
the plot. (**C**) Agarose gel electrophoresis of PCR products amplified from DNA of subclones which were derived for
gated cells as indicated on top of the gel image. The primer pairs used for the amplifications are shown above the
lanes. The lower scheme shows the positions of the primers and the expected sizes of the PCR products for the
transfected construct and its recombinants.

ATG start codon upstream of the *GFP* coding sequence (**Figure 7D**). This change—explaining the
strongly increased green fluorescence of R6 cells—was most likely enabled by a stretch of 12 identical
nucleotides at the beginning of the *tdT* and *GFP* coding sequences.

These results show that the appearance of R5 cells can be explained by *dHS*-templated gene con-
version leading to the loss of the first ATG codon in *uHS* whereas the appearance of R6 cells reflect
*uHS*-templated gene conversion leading to the gain of the second ATG codon at the very end of *dHS*.

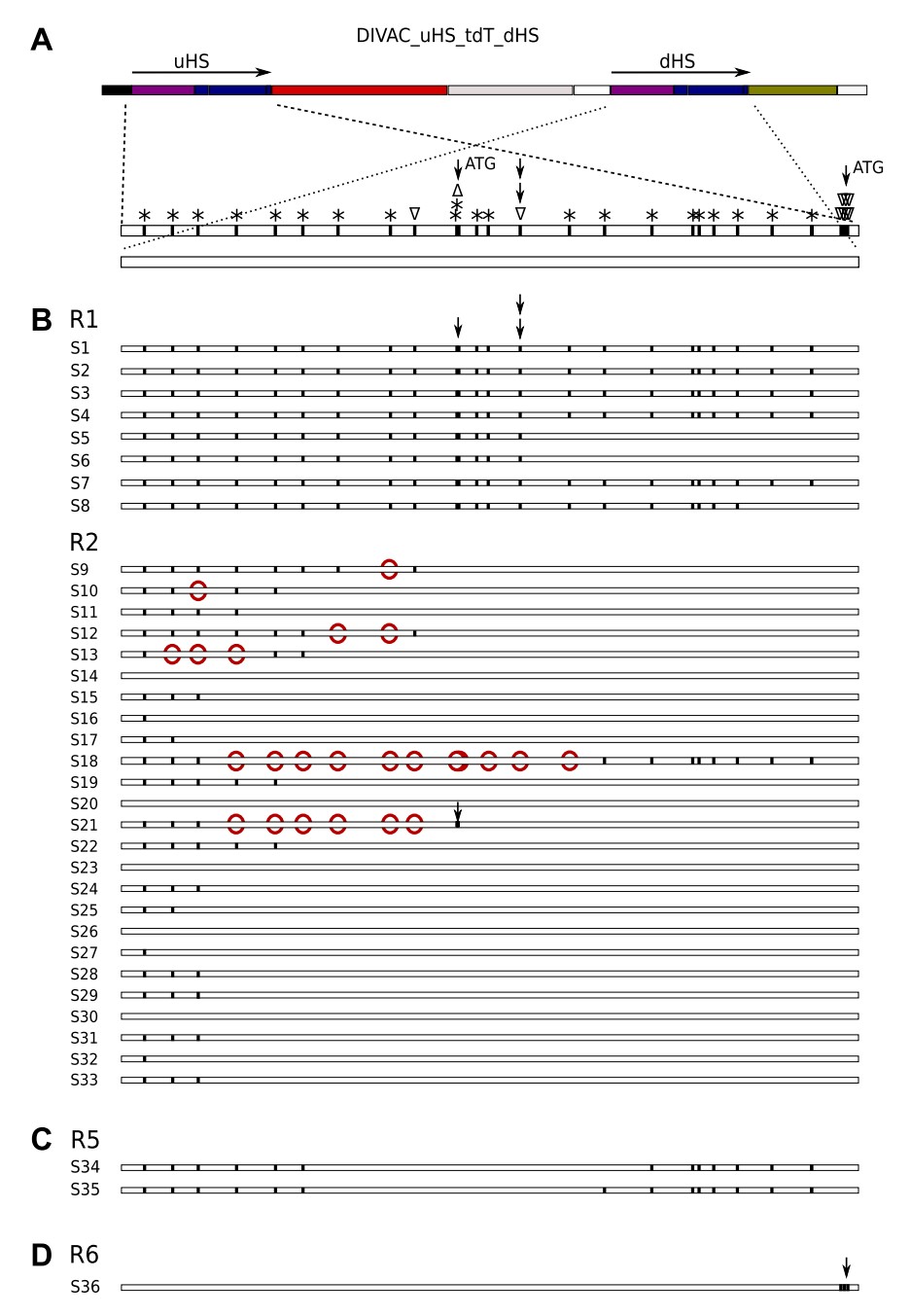

**Figure 7**. Sequences derived from gated cells of the DIVAC_uHS_tdT_dHS transfectant. (**A**) Map of the construct showing the polymorphisms of the aligned *uHS* and *dHS* sequences. The types of polymorphism are coded above the *uHS* sequence. Asterisks indicate nucleotide substitutions while triangles pointing up and down indicate single nucleotide insertions and deletions, respectively, within *uHS*. The positions of the first and second ATG start codon of *uHS* are highlighted by single arrows, and the position of the single nucleotide deletion that puts the first ATG in frame with *tdT* is indicated by stacked arrows. In all parts of the figure vertical lines within the sequence bars indicate *uHS* specific sequence. (**B**) Schematic representation of the sequences of *uHS/dHS* recombinants amplified from sorted R1 and R2 cells. *uHS* specific nucleotides downstream of *dHS* nucleotides are marked by red circles. (**C**) *uHS* sequences amplified from sorted R5 cells. (**D**) A *dHS* sequence amplified from a R6 subclone.

Gene conversions within the *uHS/dHS* repeat occurred about 3–10 times less frequently than deletions by RR, based on the analysis of subclones of three independent DIVAC_uHS_tdT_dHS transfectants (compare the medians of R1 and R2 vs R5 and R6 events in *Figure 5C*).

## Recombination of repeats accompanies switch recombination in CH12 cells

To determine whether RR can accompany CSR in chicken DT40 and murine CH12 cells, we designed the DIVAC_RSV_Sμ_tdT_RSV_Sα construct in which both the *tdT* and the *GFP* genes were driven by *RSV* promoters and switch region sequences were placed within intron sequences (*Figure 8A*). In this construct, CSR between the two S regions should generate R1 cells (high GFP due to the start codon upstream of the recombined S regions), whereas RR of the *RSV* promoter regions should generate R2 cells (low GFP), as described above.

When DIVAC_RSV_Sμ_tdT_RSV_Sα was introduced into DT40 by targeted integration, transfectants accumulated cells in the R1 and R2 gates at median frequencies of about 0.15% and 5% respectively after 12 days culture (*Figure 8B1* and data not shown). Transfectants of CH12 varied in their FACS profiles most likely due to integration of the transfected construct at variable chromosomal positions. However, about one in three transfectants generated a low number of cells similar to R1 and R2 cells of the DT40 transfectants. The frequency at which R1 and R2 cells were generated was strongly increased when factors known to increase both AID expression and *Ig* CSR (*Kinoshita et al., 1998*) were added to the culture of CH12 transfectants (compare *Figure 8B2* to *Figure 8B3* and data not shown). Whereas the DT40 transfectant showed about 50 times fewer R1 than R2 cells, the frequency of cells in the two gates was roughly equal in CH12 transfectants indicating increased CSR in CH12 as compared to DT40 cells. CH12 transfectants also generated strong subpopulations of R3 cells (*Figure 8B3*), but almost all of these cells disappeared when blasticidin was added to the cultures (data not shown) suggesting that these cells arose through transcriptional silencing of the *tdT* gene and not inactivation by hypermutation.

R1 cells of a DT40 transfectant as well as R1 and R2 cells of a stimulated CH12 transfectant (*Figure 8C*) were sorted yielding the cell populations DT40 R1, CH12 R1, and CH12 R2, respectively. DNA from the sorted populations as well as from the unsorted CH12 transfectant were amplified by the 1/5 primer pair. Amplification from DT40 R1 (*Figure 8D*, lane 1) and CH12 R1 (*Figure 8D*, lane 3) cells produced a smear of fragments in the range of about 2.5–3.2 kb as expected for diverse switch region recombination events. In contrast, a discrete fragment of about 3.1 kb was amplified from CH12 R2 cells (*Figure 8D*, lane 4) and to a lesser degree from CH12 unsorted cells (*Figure 8D*, lane 2) as expected for RSV repeat recombination.

Fragments amplified from DT40 R1 and CH12 R1 cells were subcloned and sequenced. Sequences from both DT40 and CH12 cells had joined the upstream *Sμ* to downstream *Sα* switch regions at variable positions displaying unusually long junctional microhomologies (*Figure 9A,B*) similar to recombination events previously reported using similar constructs (*Kinoshita et al., 1998*; *Okazaki et al., 2002*). To confirm recombination of *RSV* repeats in CH12 cells, we subcloned and sequenced the 3.1 kb band amplified from sorted CH12 R2 DNA. Ten sequences showed uniform deletions of one *RSV* repeat and the intervening sequence with no other sequence changes as expected for faithful RR events. These results demonstrate that DT40 and CH12 cells carried out both *RSV* and switch region recombination, but the absolute and relative frequencies of switch recombination were about 5 and 30-fold higher, respectively, in induced CH12 cells.

## Discussion

The finding that AID can induce chromosomal deletions by homologous recombination of repeats (RR) adds another activity to the remarkable array of AID-induced mutation and recombination events. RR mediated deletions could be visualized after chromosomal integration of fluorescent reporter constructs by virtue of changes in the fluorescence profiles of individual cells within expanding cultures. Deletions occurred at high frequencies in different sequence contexts and in both the chicken DT40 and the murine CH12 cell lines. Interestingly, transfectants of DT40 recombined the duplicated *RSV* promoters and *tdT* repeats as frequently by RR as they inactivated the RFP genes by somatic hypermutation (SH). Although DT40 diversifies its *Ig* genes by gene conversion (GC), an artificial homeologous sequence upstream of the *tdT* and *GFP* genes was recombined more often by RR than it was modified by GC. Similarly, class switch recombination (CSR) active CH12 cells

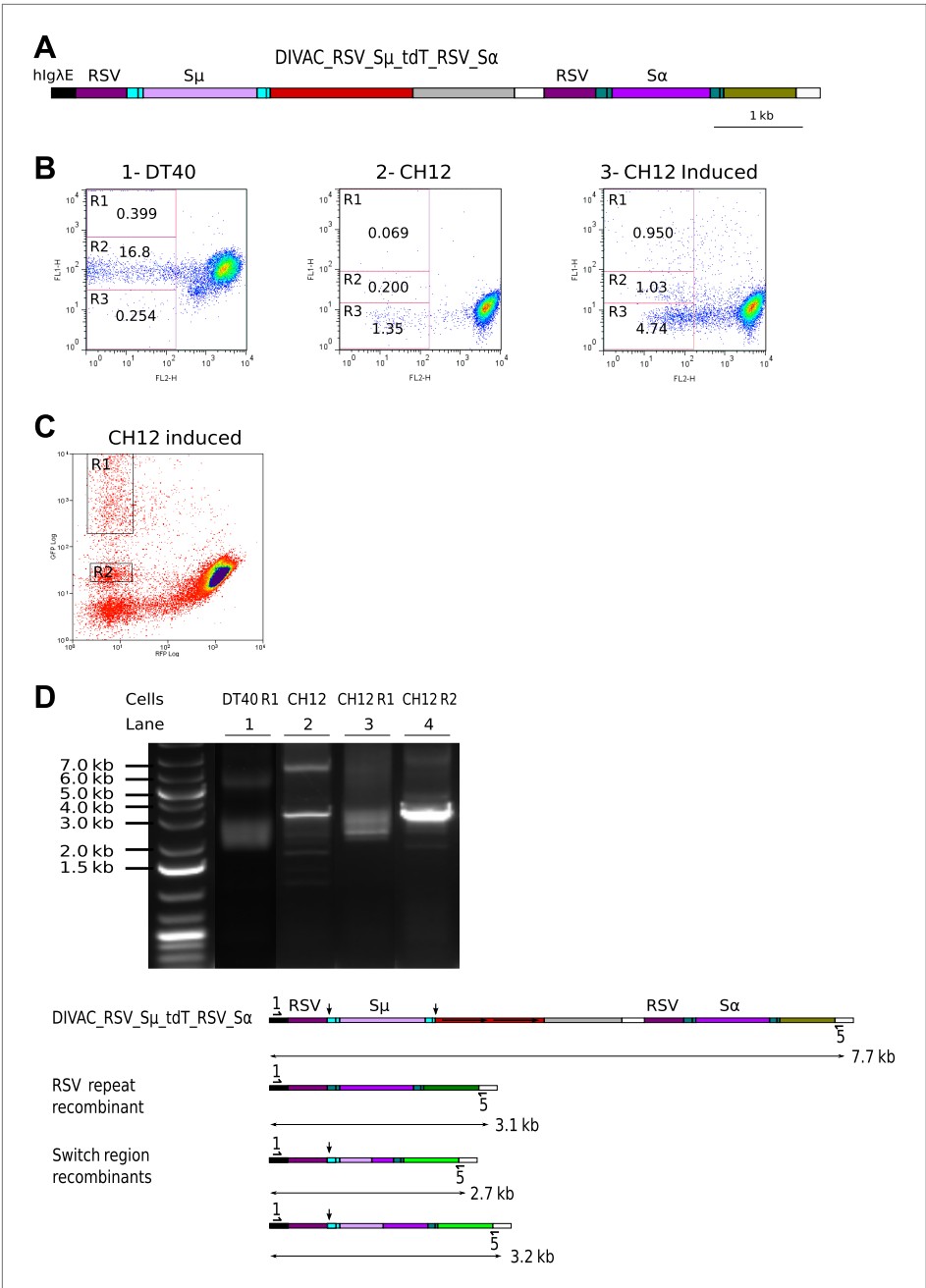

**Figure 8**. *RSV* repeat and class switch recombination after transfection of CH12 cells. (**A**) Diagram of the CH12 DIVAC_RSV_Sμ_tdT_RSV_Sα construct. *Sμ* and *Sα*—portions of the murine *Sμ* and *Sα* switch regions. (**B**) FACS profile of a representative DT40 transfectant as well as an uninduced and induced CH12 transfectant. (**C**) FACS profile of the induced CH12 transfectant showing the gates of the preparative sorts. (**D**) Top: agarose gel electrophoresis of PCR products amplified by the 1/5 primer pair from DNA of sorted DT40 R1 cells, the induced CH12 transfectant and sorted CH12 R1 and CH12 R2 cells. Bottom: the diagrams show the positions of the primers and the expected sizes of the PCR products for the transfected construct and its recombinants.

rearranged a construct containing duplicated *RSV* promoters and intronic switch regions as often by RR as by CSR. AID-induced RR does not seem to involve error prone DNA synthesis because precise deletions, but no non-templated nucleotide changes, were encountered in a large number of recombinant sequences. Our data indicate that AID-mediated RR can be a remarkably efficient process.

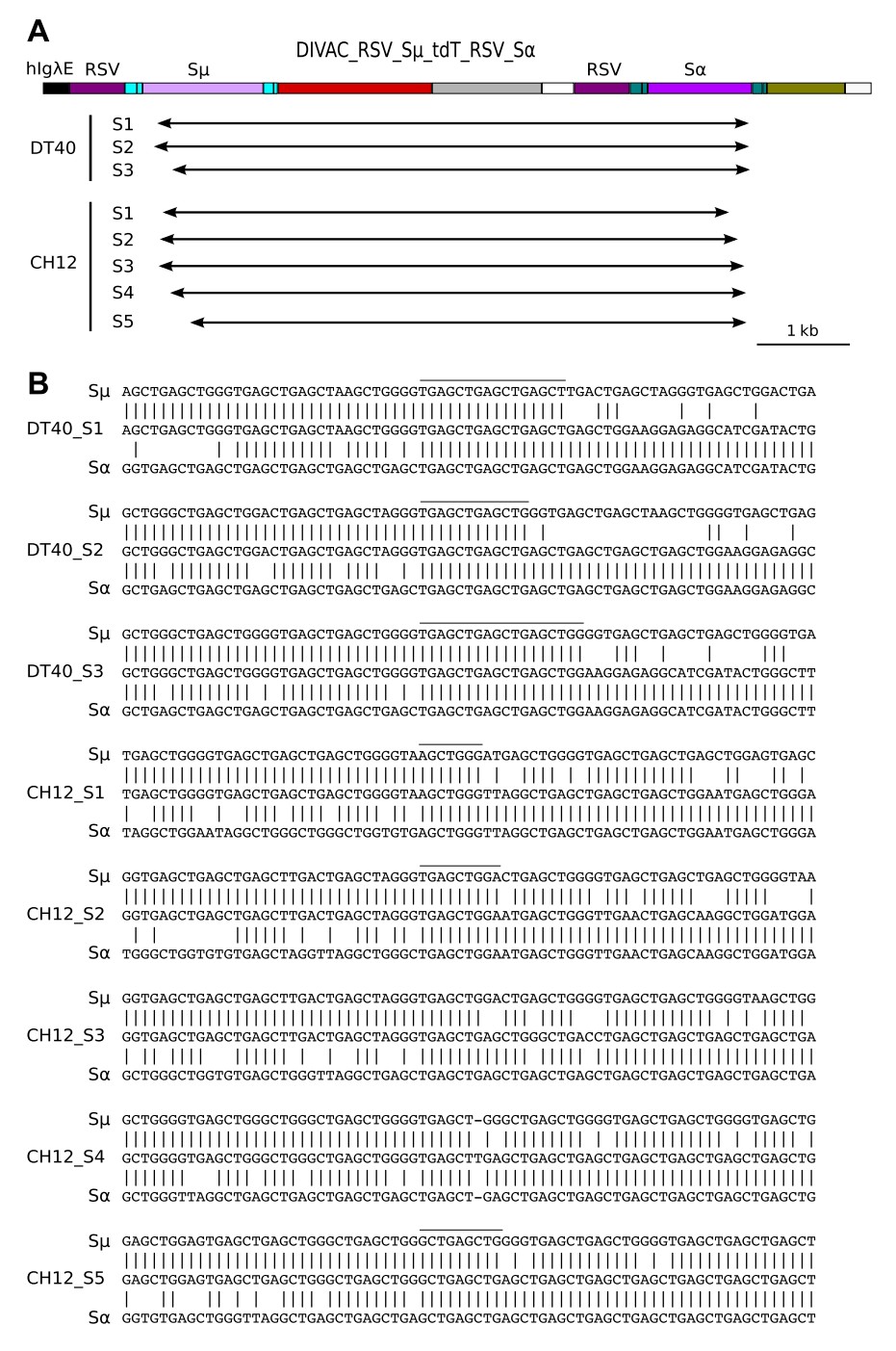

**Figure 9**. Sequences of switch region recombinants recovered from sorted DT40 R1 and CH12 R1 cells. (**A**) Deletions induced by the joining of *Sμ* and *Sα* switch regions are indicated below in the map of the DIVAC_RSV_Sμ_tdT_RSV_Sα construct by horizontal arrows. (**B**) Switch region junctions are aligned to the sequence of *Sμ* and *Sα*. Aligned *Sμ* and *Sα* sequences of the construct are shown above and below, respectively, the sequence of each switch recombinant. Junctional microhomologies are marked by a line above the sequence.

To explain RR, we propose that the single strand nick—believed to occur after the excision of an AID-induced uracil (***Figure 10B***)—is further processed into a homologous recombination intermediate. This could either be a single strand gap generated by nuclease mediated resection of the nicked strand (Model 1, ***Figure 10C–F***) or a double strand break believed to occur when the replication fork

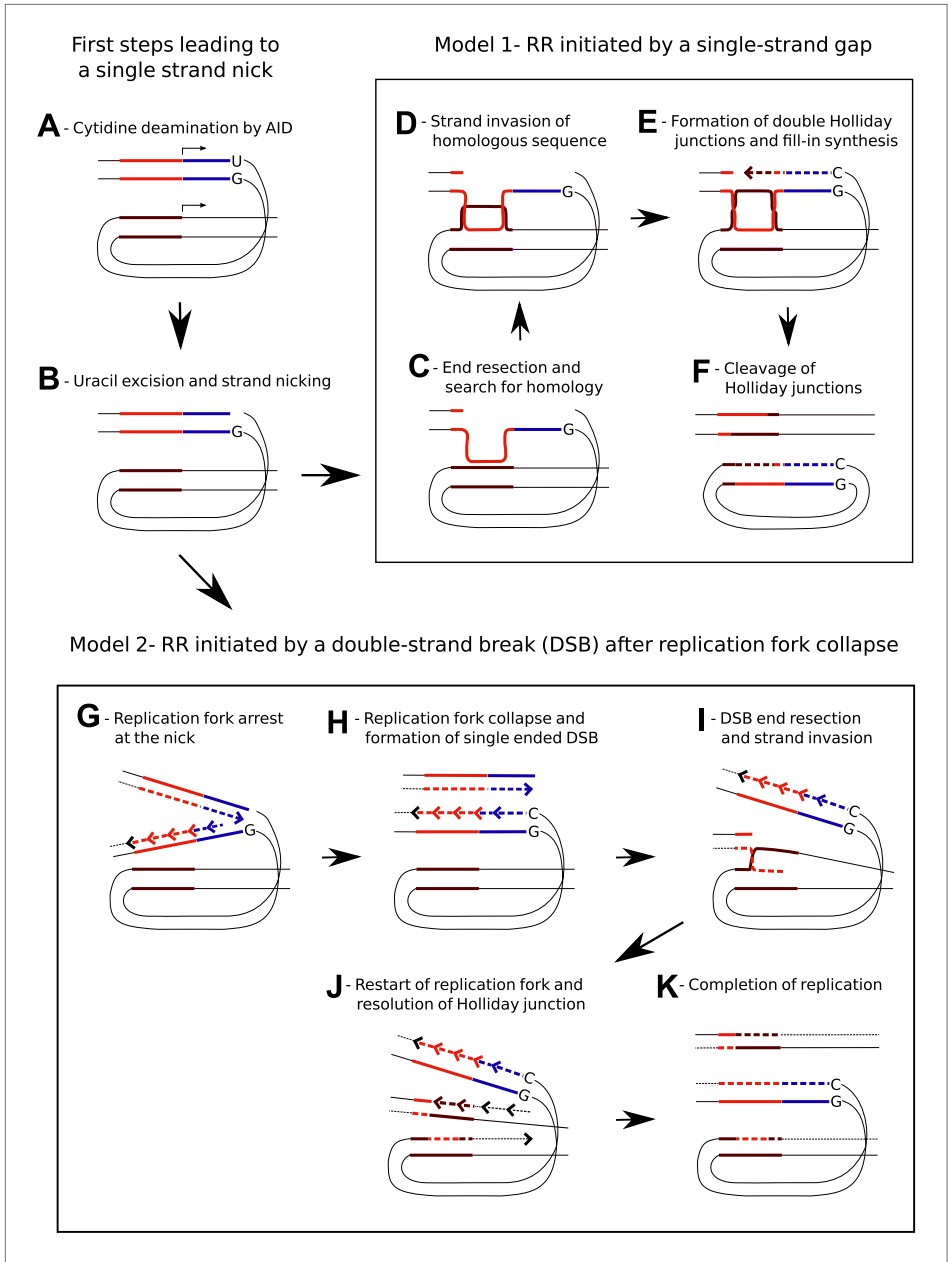

**Figure 10**. Models for AID induced Recombination of Repeats (RR) assuming initiation of homologous recombination by a single strand gap or a double strand break. (**A**) The diagrams show two neighboring genes containing direct sequence repeats marked in red and brown. The transcription start sites and the direction of transcription are indicated by horizontal arrows. Deamination of a cytidine by AID within the transcribed sequence of the first gene, marked in blue, leads to an uracil/guanidine base pair ('U' opposite to 'G'). (**B**) Removal of the uracil and cleavage of the abasic site results in a single-strand nick that is postulated to be a common intermediate for the two models. (**C**) The first model (Model 1, **C–F**) assumes that 5' to 3' resection of the nick produces a gapped DNA duplex and that the unpaired, continuous strand initiates the search for homology. (**D**) D-loop formation at the downstream repeat sequence. (**E**) Following strand exchange DNA synthesis shown by the dashed lines fills in the gap in the DNA duplex. (**F**) Cleavage of the Holliday junctions in one plane creates a chromosome containing a deletion and a circular DNA molecule of the deleted sequence. Cleavage of the Holliday junctions in the other plane would result in a chromosome with no deletion (not shown). (**G**) The second model (Model 2, **G–K**) assumes replication arrest at the AID-induced nick. (**H**) The nick is converted into a single ended DSB due to replication fork collapse. (**I**) Upon resection the DSB erroneously re-initiates the replication fork at the position of the downstream repeat sequence. (**J**) The Holliday junction is cleaved and replication continues. (**K**) Different sister chromatids, one carrying a deletion and one without a deletion, are produced by completion of replication.

collapses at the site of the unrepaired single strand nick within one of the template strands (*Kowalczykowski, 2000*; *Petermann and Helleday, 2010*) (Model 2, *Figure 10G–K*). Since recombination occurred at high frequency between the *RSV* repeats, which are upstream of the transcription start site and hence likely to be hundreds of bases away from the peak of AID-induced cytidine deamination (*Saribasak et al., 2006*), both models invoke substantial DNA resection to expose single stranded DNA distant from the nicks (*Figure 10C,I*). The fact that RPA (*Hakim et al., 2012*) as well as RPA together with RAD51 (*Yamane et al., 2013*) accumulated at multiple chromosomal positions within activated murine B cells in an AID-dependent fashion is consistent with the formation of recombination competent nucleoprotein filaments. Later stages of the models postulate heteroduxplex DNA and Holliday junctions (*Figure 10E,I*), which are consistent with the sequence analysis of homeologous repeat recombinants (*Figure 7*). Nevertheless, the choice, validity, and details of the two alternative models remain speculative. Further insight into the RR reaction will likely be provided by the analysis of constructs containing inverted repeats and the behavior of constructs in cells with specific recombination and repair factor deficiencies (*Hu et al., 2013*; *Willis et al., 2014*).

RR needs to be thought about within the context of Ig locus diversification and the repair of AID-induced DNA damage. For example, it could be responsible for V gene replacements that are not associated with cryptic V(D)J signal recognition sequences, as these were observed in AID expressing germinal center B cells (*Darlow and Stott, 2005*). Similar to RR, GC, which diversifies the rearranged Ig genes of B cells in chickens and many mammalian species (*Reynaud et al., 1987*; *Butler 1998*), requires the interaction of nearby homeologous sequences on the same chromosome (*Arakawa et al., 2004*; *Sale, 2004*). Intriguingly, the two processes might involve the same intermediates leading to sequence alignment and strand invasion (*Figure 10C,D*) and later diverge when the Holliday junctions (*Figure 10E*) are resolved by cleavage during RR, but by convergent branch migration during GC. How GC and RR are controlled during Ig repertoire development remains unresolved. Both of the fluorescent reporter genes in our constructs were transcribed and expected to accumulate AID induced nicks which might facilitate RR. On the contrary, the pseudo V genes within the chicken Ig loci are unlikely to be transcribed, and the absence of transcription or their less accessible chromatin configuration may favor GC.

While possibly involved in the repair of AID-induced DNA damage, RR is also likely to add to the mutation signature of AID by introducing deletions and inversions of repetitive sequences. AID expressing B cells are prone to transformation (*Robbiani and Nussenzweig, 2013*) and mutation signatures of APOBEC homologues are found in the genomes of many cancer cells (*Alexandrov et al., 2013*; *Burns et al., 2013*; *Taylor et al., 2013*). *Cis*-acting sequences that activate SH in nearby transcribed sequences and which have been termed DIVersification ACtivators (DIVACs) (*Buerstedde et al., 2014*), strongly stimulated the frequency of RR, revealing an additional risk to genome integrity posed by DIVAC-like sequences outside of the *Ig* loci. However, RR-mediated deletions were also detectable after transfection of 'no DIVAC' containing control constructs (e.g., R2 events arising from the RSV_tdT_RSV substrate; *Figure 2B,C*). Thus, AID and perhaps other cytidine deaminases represent a general threat to transcribed regions of the genome via multiple mechanisms including RR, as reported here. Given the highly repetitive nature of mammalian genomes, even a low level of RR could significantly contribute to genome instability and B cell transformation.

## Materials and methods

### Construction of RFP and dual fluorescence reporter constructs

The *RSV* promoter, *GFP*, *IRES-Bsr*, and *SV40* polyA sequences used for the RFP gene expression cassettes and the *hIgλE* and *hIgHiE* DIVACs had been previously described (*Buerstedde et al., 2014*). The *u-exon-intron-exon* sequence was derived from the *leader-intron-V gene* sequence of the chicken *IgL* gene. The *tdT*, *DsR*, *Ubiquitin C promoter*, *d-exon-intron-exon* sequence, and the *BGH* polyA signal were amplified from Addgene plasmids pcDNA3.1(+)/Luc2 = tdT, pDsRed-Sensor, pUB-GFP, pCI-FlagPCAF, and pcDNA3.1(+)/Luc2 = tdT, respectively. The *iHS* and the *uHS* sequence was custom synthesized. The *dHS* sequence was identical to the *RSV* promoter and the *d-exon-intron-exon* sequence. The partial *Sμ* and *Sα* switch regions of about 1.3 kb and 1.2 kb size were amplified from the plasmid *SCI(μ,α)* (*Okazaki et al., 2002*). All cloning steps, sometimes including additions or deletions of restrictions sites, were done by the Infusion Cloning Kit (Clontech, Mountain View, CA) after PCR amplifications of fragments using Q5 High-Fidelity DNA Polymerase (NEB, Ipswich, MA).

## Cell culture

DT40 transfectants having integrated the fluorescence reporter constructs targeted at the position of the deleted *IgL* locus of the IgL(−) variant cell line were identified as previously described (*Blagodatski et al., 2009*). All transfectants were initially selected by blasticidin, but cultured subsequently in the absence of blasticidin. Subcloning was also performed in the absence of blasticidin. To test for blasticidin sensitivity, cultures were split and one half was cultured for 3 days in the presence of blasticidin, the other half in the absence of blasticidin.

Transfection of CH12 cells was done using a Gene Pulser Xcell (BioRad, Hercules, CA) electroporator and a square wave protocol of 230 V and 20 ms. Transfectants were initially selected in medium containing blasticidin at a concentration of 15 µg/ml. Stimulation of CH12 transfectants was performed in medium containing 5 ng/ml IL4, 0.2 µg/ml anti-CD40 antibody, and 0.1 ng/ml TGFβ for 5 days.

## FACS analysis

FACS analysis of transfectants carrying the red and dual fluorescence constructs was similar to the previously described analysis of transfectants carrying green fluorescence reporters (*Blagodatski et al., 2009*; *Buerstedde et al., 2014*). Green (FL1) and red fluorescence (FL2) were plotted on the y-axis and x-axis respectively using a FACSCalibur (BD Biosciences, San Jose, CA). Excitation was by the 488 nm laser and appropriate FL1/FL2 and FL2/FL1 settings were used for compensation. Despite likely suboptimal excitation of tdT by the 488 nm laser, the red fluorescence of tdT expressing cells was very bright, causing difficulty in displaying positive and negative cells on the same screen. FACS settings were optimized for the analysis of either tdT or DsR expressing cells and then consistently used for the analysis of respective primary transfectants and subclones. Primary transfectants were analyzed 20 days after transfection (about 12 days in blasticidin free culture medium), subclones were analyzed 12 days after subcloning. Over 20 subclones of each transfectant were usually analyzed. Preparative FACS sorts were performed on Beckman Coulter MoFlo using the following parameters: 488 nm excitation, 100 mW power laser for GFP; 532 nm, 200 mW laser for RFP expression.

## PCR and sequence analysis

PCR amplifications were performed using Q5 High-Fidelity DNA Polymerase (NEB, Ipswich, MA) and genomic DNA isolated either from primary transfectants, from sorted cell populations or from subclones. PCR reactions were analyzed by electrophoresis on 0.8% agarose gels.

For sequence analysis, PCR fragments were excised from the agarose gels and cloned into the linearized pUC19 provided with In-Fusion Cloning Kit (Clontech, Mountain View, CA). Sequences of the subcloned PCR fragments were compared to the sequence of the transfected construct using 'Align two sequences' blastn searches (http://blast.ncbi.nlm.nih.gov/).

## Acknowledgements

We would like to acknowledge Gouzel Tokmoulina and Cameron Shaw Godecke for their expert assistance with the preparative FACS sorts and Julian Sale, Ciaran Morisson, and Hiroshi Arakawa for helpful comments on the manuscript. We particularly thank the eLife reviewers and Wolf-Dieter Heyer for advice on mechanisms of recombination. JMB was supported by an International Outgoing Marie Curie Fellowship. DGS is an investigator of the Howard Hughes Medical Institute.

## Additional information

### Funding

| Funder | Grant reference number | Author |
| --- | --- | --- |
| European Commission | Marie Curie International Outgoing Fellowship | Jean-Marie Buerstedde |
| Howard Hughes Medical Institute | | David G Schatz |

The funders had no role in study design, data collection and interpretation, or the decision to submit the work for publication.

## Author contributions
J-MB, Conception and design, Acquisition of data, Analysis and interpretation of data, Drafting or revising the article, Contributed unpublished essential data or reagents; NL, Analysis and interpretation of data, Drafting or revising the article; DGS, Conception and design, Analysis and interpretation of data, Drafting or revising the article

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
