## [Decision Letter]

Thank you for sending your work entitled “Induction of reciprocal homologous recombination by AID” for consideration at *eLife*. Your article has been favorably evaluated by James Manley (Senior editor), a Reviewing editor, and two reviewers.

The Reviewing editor and the two reviewers discussed their comments before reaching this decision, and the Reviewing editor has assembled the following comments to help you prepare a revised submission.

The first author of this manuscript has previously identified a unique enhancer, DIVAC, which increases activated-deaminase (AID)-dependent mutagenesis (PLOS Biol 2014). The current manuscript uncovers another and unexpected function of AID as well as for DIVAC. The work here shows that they collaboratively promote reciprocal recombination between two homologous sequences. Remarkably, they are very frequently subjected to reciprocal recombination, even though the two homologous sequences are relatively short, only up to 1kb, and localized at a distance of several-kb. The finding significantly contributes to the understanding of AID-mediated mutagenesis involving repeat sequences. The authors carefully designed experiments, did an impressive amount of work, and confirmed reproducibility of data using various recombinant plasmids. We therefore recommend publication in a high profile journal such as *eLife*, if appropriate revision is done, particularly on the model figure.

Our major concern is the model for reciprocal recombination shown in Figure 10. The authors propose that single-strand breaks (step 4) triggers end-resection (step 5) and subsequent strand invasion into homologous sequences (step 6). Given the extremely quick repair of single-strand breaks in mammalian cells, it seems unclear that a single-strand breaks could be efficiently resected and stimulate reciprocal recombination. Another possibility is that abasic sites (step 3) stall DNA replication: replication stalling may be released by template switching to adjacent homologous sequences. The manuscript should be published soon if the authors discuss other possibilities for initiation of reciprocal recombination by replication blockage at AID-induced DNA damage sites. A balanced model might consider the one proposed and others. In any case, if the authors want to stick with only their own speculative ideas they must address the following.

As the authors referenced, there is some evidence that HR may be initiated by an SSB as well as by double-strand breaks (DSBs), although the mechanism has yet to be worked out. In Figure 10(5), what enzyme would resect a nick and, if the nick is resected, what is left over to carry out the homology search? That is only one strand would make a D-loop upon invasion and this would be an unusual structure to resolve. It may be that a helicase would be a more likely player. Also, for a large single-strand gap initiated by a SSB, the complementary strand is still intact, so what would prevent DNA ligase from filling in the single-strand gap instead of the cell initiating a homology search? In Figure 10(7), most models I have seen for double Holliday require second-strand invasion, which cannot occur if you only have a SSB. The events observed seem like they may still be accounted for by conversion of a SSB to a DSB, perhaps during replication over the nick leading to fork collapse. If this occurred, repair may be favored by HR due to the cell being in S-phase and may preferentially repair with nearby sequences due to the large number of AID-initiated nicks, since nicks have been shown to stimulate repair of DSBs.

Minor comments:

1) The authors define reciprocal homologous recombination events as GC without cross over. The word “reciprocal” connotes an exchange of genetic data with no net loss whereas most of the events described are non-conservative deletions.

2) In the graphs of the FACS data in Figures 1, 2, 3 and 5, is it possible to perform an analysis (t-test, Mann-Whitney) on the data to determine if there is statistical significance?

3) In the text, it is stated that the “two ATG” cassette was added for the Figure 2 substrate, but it is also shown in the Figure 1 substrate. Is it in both?

4) In Figure 2(2) in appears that some of the intermediate red population is overlapping with the R3 gated population.

5) Since this is a study on recombination, it is unclear why the authors chose to continue using the tdT reporter with a direct repeat when the DsR appears to work just as well without the added variable of deletion events. Especially since the 2nd population of supposed single RFP seems quite sizable in the FACS dot blots. The results may lend to easier interpretation with the DsR construct.

6) The presence of the blasticidin seems like it would enable the authors to determine the frequency of deletion events. While this would limit the number of detectable events compared to FACS, it could complement the FACS data and report the number of viable cells that survive deletions.

7) In Figure 4 the primary transfectant (Lane 1-4) produces both the 2.5 and 1.7 kb fragment for the 1/3 pair and the major products for the 1/4 and 1/5 pair seem to correspond to a deletion of the cassette. Is the cassette that unstable or could it be that some of these are PCR products are artifacts? For example, a downstream RSV amplified by primer 4 could anneal with the upstream RSV amplified by primer 1 leading to the amplification of a smaller product. This could be an issue in Figure 6 as well. Southern blots may be the only way to absolutely confirm these proposed recombination products.

8) How long are the Smu and Salpha sequences described in Figure 8?

9) The fact that the cassette was randomly integrated into CH12 cells and that the results were variable makes it difficult to interpret the results of this experiment. While HR can occur in CH12 cells, CSR has shown to occur by NHEJ. That the experiments are not performed in the context of the switch region using switch promoters also make the interpretation and biological significance difficult to interpret.

10) Could loss of R3 cells and blasticidin resistance in some of the CH12 cells also be explained by NHEJ that joins the DNA so that the genes are not in frame with their start codons?

11) For Figure 8 switch regions are difficult to amplify through, even with a high-fidelity polymerase, so Southern blots may again be the best way to confirm the proposed substrates.

---

## [Author Response]

*Our major concern is the model for reciprocal recombination shown in*
Figure 10*. The authors propose that single-strand breaks (step 4) triggers end-resection (step 5) and subsequent strand invasion into homologous sequences (step 6). Given the extremely quick repair of single-strand breaks in mammalian cells, it seems unclear that a single-strand breaks could be efficiently resected and stimulate reciprocal recombination. Another possibility is that abasic sites (step 3) stall DNA replication: replication stalling may be released by template switching to adjacent homologous sequences. The manuscript should be published soon if the authors discuss other possibilities for initiation of reciprocal recombination by replication blockage at AID-induced DNA damage sites. A balanced model might consider the one proposed and others. In any case if the authors want to stick with only their own speculative ideas they must address the following*.

*As the authors referenced, there is some evidence that HR may be initiated by an SSB as well as by double-strand breaks (DSBs), although the mechanism has yet to be worked out. In*
Figure 10*(5), what enzyme would resect a nick and, if the nick is resected, what is left over to carry out the homology search? That is only one strand would make a D-loop upon invasion and this would be an unusual structure to resolve. It may be that a helicase would be a more likely player. Also, for a large single-strand gap initiated by a SSB, the complementary strand is still intact, so what would prevent DNA ligase from filling in the single-strand gap instead of the cell initiating a homology search? In*
Figure 10*(7), most models I have seen for double Holliday require second-strand invasion, which cannot occur if you only have a SSB. The events observed seem like they may still be accounted for by conversion of a SSB to a DSB, perhaps during replication over the nick leading to fork collapse. If this occurred, repair may be favored by HR due to the cell being in S-phase and may preferentially repair with nearby sequences due to the large number of AID-initiated nicks, since nicks have been shown to stimulate repair of DSBs*.

Thank you for the fast and very helpful evaluation of our manuscript. Based on the comments of the reviewers and editors, we have substantially revised the manuscript and the figures. The one major criticism of the manuscript related to our model (Figure 10) for how AID induces deletion between repeats by homologous recombination. The comments were insightful and prompted us to revise the single strand nick/gap version of the model (so that now, initial strand invasion is mediated by the intact DNA strand) and to add an alternative, replication-induced double strand break version of the model. We have also rewritten the paragraph of the Discussion dealing with the models. We believe that it will be valuable to present both models given the uncertainty of how these AID-induced events occur: as noted in the review “there is some evidence that HR may be initiated by an SSB as well as by double-strand breaks (DSBs)”.

In response to the minor comments we have:

- changed the term ‘Reciprocal Homologous Recombination’ into ‘Homologous Recombination of sequence Repeats’ (leading to a change in the title of the manuscript and further changes throughout the manuscript)

- added information about the position of the ATG start codons for the constructs shown in Figure 1

- added the sizes of the switch regions

- simplified and better explained the naming of the constructs.

We share the view that PCR products of repetitive sequences can be difficult to interpret. However, since the identity for the bands, thought to represent the different types recombinant sequences, was verified by subcloning and sequencing, we feel that Southern blot analyses would add little and unduly delay publication of the manuscript. We also note that a preponderance of smaller PCR products indicative of deletion (such as those with pairs 1/4 and 1/5) could be due largely or entirely to preferential amplification of smaller PCR products at the expense of the full-length products. We believe that the combination of PCR, sequencing, and FACS data provides a clear picture of the frequency of the various events, and that a more detailed analysis taking advantage of blasticidin resistance would not substantially add to this.

Also, since our results, relying on the medians of large numbers of subclones from multiple independent transfectants and often multiple similar constructs, show clear-cut differences to support the main conclusions of the study, we have not added further statistical analysis.

Regarding the CH12 experiments, they were not designed to provide an analysis of class switch recombination (as was done by others some years ago using similar switch substrates) but rather to test the hypothesis that AID could also induce homologous recombination of sequence repeats in a mouse cell line that differed considerably from the chicken DT40 line. Inclusion of switch sequences provided a useful internal control and point of reference for interpreting the homology directed deletions.

Finally, regarding our use of tdT-containing constructs (instead of DsR) for most of the experiments, we do not feel that this choice limits any of the conclusions, and it certainly emphasizes the abundance of AID-mediated homologous recombination between sequence repeats.